# Olfactory Critical Periods: How Odor Exposure Shapes the Developing Brain in Mice and Flies

**DOI:** 10.3390/biology13020094

**Published:** 2024-02-02

**Authors:** Ahana Mallick, Andrew M. Dacks, Quentin Gaudry

**Affiliations:** 1Department of Biology, University of Maryland, College Park, MD 20742, USA; amallick@umd.edu; 2Department of Biology, West Virginia University, Morgantown, WV 26505, USA; andrew.dacks@mail.wvu.edu

**Keywords:** critical period, olfactory circuit, fruit flies, mice, neuromodulators

## Abstract

**Simple Summary:**

Critical periods have been extensively studied in the context of the visual system in mammals. Despite an immense interest in critical periods, much less is known about the cellular and molecular mechanisms involved in olfactory critical periods. This review provides an overview of the olfactory critical periods drawing from the literature on both mice and fruit flies. We draw parallels to cellular and molecular mechanisms identified in the visual system to guide our discussion on critical periods in the olfactory circuit.

**Abstract:**

Neural networks have an extensive ability to change in response to environmental stimuli. This flexibility peaks during restricted windows of time early in life called critical periods. The ubiquitous occurrence of this form of plasticity across sensory modalities and phyla speaks to the importance of critical periods for proper neural development and function. Extensive investigation into visual critical periods has advanced our knowledge of the molecular events and key processes that underlie the impact of early-life experience on neuronal plasticity. However, despite the importance of olfaction for the overall survival of an organism, the cellular and molecular basis of olfactory critical periods have not garnered extensive study compared to visual critical periods. Recent work providing a comprehensive mapping of the highly organized olfactory neuropil and its development has in turn attracted a growing interest in how these circuits undergo plasticity during critical periods. Here, we perform a comparative review of olfactory critical periods in fruit flies and mice to provide novel insight into the importance of early odor exposure in shaping neural circuits and highlighting mechanisms found across sensory modalities.

## 1. Introduction

Experience-dependent neuronal plasticity is one of the hallmarks of the nervous system. Exposure to environmental stimuli induces heightened levels of circuit refinement and plasticity in response to the stimuli. In the early postnatal life of an organism, there exists a specific time window when neuronal circuits show heightened levels of experience-dependent plasticity. This time of heightened neuronal plasticity is called the critical period and was first defined by the Nobel prize-winning work of Hubel and Wiesel in 1962 in the context of the development of cortical receptive fields of binocular vision [1,2,3,4,5]. Since then, critical periods have been discovered in multiple sensory modalities across species [6,7,8], including the visual, auditory, somatosensory, and olfactory cortices of mice [9,10,11,12] as well as the visual and sensorimotor circuits [13,14] of the fruit fly larvae and in the olfactory system of adult flies [15,16,17,18,19,20,21,22,23,24,25,26,27,28,29,30,31,32,33]. Experience-dependent plasticity can occur both during and after the critical period. In this review, we focus on experience-dependent plasticity during the critical period which we will refer to as critical period plasticity (CPP), which exhibits the following features as reviewed in Sengpiel 2007, Cioni and Sgandurra 2013, and Knudsen 2004 [34,35,36]: 1.CPP can only be induced at a specific time window in the early life of an organism in response to repeated experience-dependent activity in the circuit and result in stable differences in physiology and/or behavior [34,35,36] (Figure 1).2.It occurs when the sensory circuits are still developing but have achieved reliable and precise inputs [36].3.In addition to the presence of excitatory components, CPP onset is marked by the appearance/arrival of the inhibitory components in the circuit [34].4.During CPP, changes occur both at the level of synaptic transmission and structure induced by activation of gene transcription and translation that ultimately lead to long-term functional changes [36].

**Figure 1 biology-13-00094-f001:**
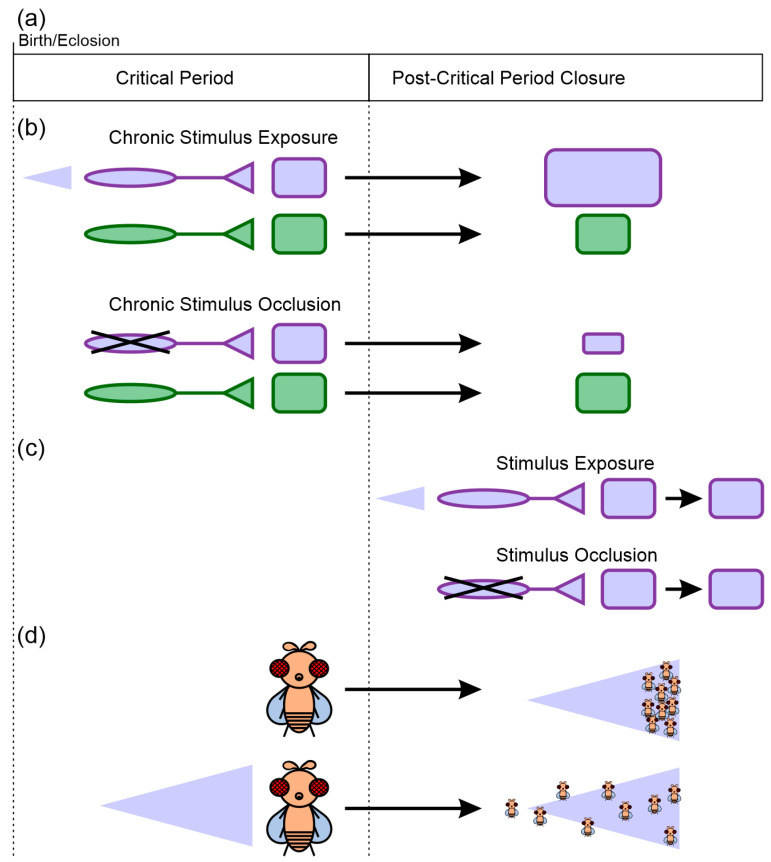
Core features of critical periods. (**a**) Critical periods are windows of time during which exposure to specific environmental stimuli can induce changes in the nervous system’s structure and function. Once the critical period closes, the capability for plasticity is greatly reduced. (**b**) Critical period plasticity is stimulus-specific. Chronic stimulus exposure or occlusion during the critical period can induce changes in nervous system architecture and function selectively for those regions that process the stimulus (purple square), but not necessarily for other regions (green square). The purple triangle represents exposure to an odor that only activates some sensory afferents (purple) but not others (green). (**c**) Equivalent stimulus exposure after the closure of the critical period does not induce changes equivalent to those induced during the critical period. (**d**) The neural plasticity induced during the critical period can result in sustained differences in behavioral preferences. In this example, naïve flies distribute and move to high odor concentrations (right side of the purple triangle), while flies that experience chronic exposure to the odor do not orient themselves based on a concentration gradient for that specific odor.

Experience can shape neuronal circuits at any time during the life of an organism. For example, repeated exposure to the same odor stimulus improves the robustness and discriminability of the odor responses in fruit flies and mice [37,38,39,40,41,42]. However, neuronal circuits are more susceptible to experience-dependent neuronal plasticity during the critical period [43]. Odor imprinting is the most evocative illustration of the importance of the critical period in the olfactory system. For example, odor exposure during the olfactory critical period in neonatal mice increases its sensitivity to the odor as an adult [44,45]. Similar examples of olfactory imprinting also exist in *C. elegans* [46,47]. However, the most iconic example of olfactory imprinting memory is the honing migrations of adult salmon and trout [48,49]. Salmon develops through their embryonic to juvenile stages in fresh water and return to the same freshwater stream as adults for spawning. It is believed that juvenile salmon imprints on the odors present in the freshwater stream right before leaving the stream [50,51], which helps it to navigate back towards the same stream as adults [52,53]. This form of odor imprinting occurs at various stages of development in different salmon species and starts as early as in the embryo [54]. These examples underscore the importance of circuit refinement unique to the chemical environment to which the organism is exposed. For example, repeated exposure to the same odor stimulus improves the robustness of the odor responses and discriminability in fruit flies and mice [37,38,39,40,41,42]. However, despite the importance of the critical period in olfactory plasticity and refinement of circuits, the cellular and molecular mechanisms underlying the olfactory critical period have not been as extensively studied as visual critical periods.

Critical periods have been well-documented and extensively reviewed for multiple sensory systems and across many species. The goal of this review is to identify key principles emerging from the study of olfactory critical periods in mice and fruit flies. Therefore, we begin by providing a brief introduction to the mouse and fruit fly olfactory systems and discuss how plasticity affects these circuits during the critical period. We highlight both unique and shared morphological, functional, and behavioral features of olfactory critical periods and draw from the literature on critical periods of other sensory systems to discuss common emerging principles of sensory circuit refinement during this period. Finally, we discuss the transferability of the observations about experience-dependent critical period plasticity made in the laboratory to more naturally occurring conditions. Through a deeper understanding of how olfactory critical periods ultimately shape neuronal circuits, we can gain insight into how the external environment and experience-dependent plasticity contribute to the overall development of the olfactory model system. 

## 2. Organization of the Olfactory Processing Centers 

The olfactory system plays a crucial role in the survival of animals. It provides vital cues about the chemical environment that allows an organism to optimize feeding, reproduction, predator avoidance, and social conduct. The organization of the primary olfactory processing networks (See Figure 2b) therefore share several common themes across species [55]. In both mice and flies, odor information is received at the periphery by olfactory sensory neurons (OSNs) that express chemoreceptive proteins activated by different volatile chemicals. OSNs expressing the same complement of chemoreceptive proteins project into a primary olfactory processing center composed of discrete neuropil structures called glomeruli. OSNs activate downstream neurons within their cognate glomerulus, thereby converting an odor signal into an odor map [56,57,58,59,60,61,62,63] within the primary olfactory processing center that provides vital information about the type, chemical nature, concentration, duration, and directionality of the odor stimuli. Information from each glomerulus is further transmitted to various higher-order olfactory processing centers by second-order neurons that ultimately give rise to olfactory perception. Nevertheless, there are striking differences in the peripheral anatomy, evolution, and signaling mechanisms between the mice and *Drosophila* olfactory systems as described below. 

### 2.1. Primary Olfactory Circuit of Mice

The olfactory perception in mice begins with the activation of receptors expressed by OSNs residing in the nasal cavity’s primary olfactory epithelium [56,62,64] (See Figure 2a). Each OSN extends a single dendrite to the surface of the epithelium, equipped with immotile cilia that reach out to capture odor molecules from inhaled air. These cilia have a dense expression of ORs and associated olfactory transduction machinery that convert volatile chemicals into a change in voltage [64,65]. The axons of OSNs collectively form the olfactory nerve at the periphery. Within the OB glomeruli, OSN axons synapse upon the dendrites of output neurons called M/T cells. The functional specificity of OSNs is determined by the expression of a single type of OR from a repertoire of 1000 OR subtypes [66,67]. Odor recognition in the OB occurs upon combinatorial activation of multiple ORs of varying magnitudes [56,57]. While the dendrites of the M/T cells are present within a glomerulus, the axons of the M/T cells project through the lateral olfactory tract to the higher-order olfactory processing center such as the piriform cortex and basal forebrain. 

### 2.2. Primary Olfactory Circuit in Drosophila

In *Drosophila,* the antennae and maxillary palps are covered in thousands of sensilla housing OSNs that express a diverse set of chemoreceptive proteins. In *Drosophila*, there are 72 chemosensory receptors and 4 co-receptors [68]. OSNs that express the same combination of chemosensory receptors and co-receptors project to the same glomerulus and the different types of OSNs map onto distinct glomeruli like in mice [58,59,60,69,70,71,72,73,74,75,76,77,78]. Heteromeric complexes of distinct ligand-binding chemosensory receptors colocalize with a single or multiple co-receptor(s) to form functional ion channels that activate OSNs upon odor binding [75,79,80]. The axon of the OSNs project to glomeruli in the AL where they synapse upon second-order PNs and LNs (See Figure 2c). The PNs in turn project onto higher order olfactory centers like the MB and the LH. Within this circuit, the odor specificity map for each receptor type and glomerulus in the AL is well known [59,60,61,69,70,81,82]. At odor onset, multiple types of OSNs are activated at varying magnitudes whose combined activity provides information about the nature and concentration of the odorant molecule [58,59,71,73,76,77,78].

## 3. The Critical Period

CPP is induced by sensory stimuli; therefore, studies on critical periods are carried out using manipulations involving deprivation of, or over-exposure to, a sensory stimulus (Figure 1b). However, the core definition of the critical period rests heavily upon findings from sensory deprivation experiments that examined development and plasticity in the visual cortical circuits in kittens caused by monocular deprivation during the critical period [1,2,3,5]. They showed long-term functional changes in circuit organization and response properties in the cortex without major alterations in the peripheral circuits [4,5,83]. In contrast, experience-dependent changes in the visual circuit of mice during the critical period have been observed much earlier in the circuit in the retinal ganglion cells in response to sensory deprivation as well as over-stimulation. Dark rearing (stimulus deprivation) mice during their visual critical periods show marked differences in the development of dendritic and receptive fields [84] as well as in the thickness and length of the myelin sheath of the axons [85] of retinal ganglion cells. Daily visual stimulation in the form of optomotor response stimulation during the critical period induces BDNF-mediated hyperacuity in mice [86]. 

Similar to the visual critical period in mice, olfactory CPP has been observed at the level of the OSNs in the OB and AL of mice and *Drosophila* respectively. The critical period in the olfactory circuit is described as the time during the early life of the organism when its olfactory circuits are refined in response to its odor environment. Reminiscent of the expansion of ocular dominance columns in the visual cortex during monocular deprivation [1,5] or the expansion of the dendritic receptive fields of the retinal ganglion cells following eye-opening during the critical period [84], the olfactory critical period is marked by striking changes in the volume of the glomerulus that primarily detects the odor to which the organism is exposed [15,21,45] (See Figure 2b). It is particularly interesting to note that the critical period of monocular deprivation also coincides with the emergence and maturation of local inhibitory circuits in mice and other vertebrate model systems. Owing to the anatomical simplicity and the readily available genetic tools, critical periods of olfaction in *Drosophila* and the olfactory critical period in mice have recently received increasing attention. We propose that the olfactory system is a powerful model to study critical periods because glomerulus in the AL and OB form discrete functional units where differential changes in synaptic transmission and structure in response to an odor can be easily studied in the same identified glomerulus across animals. 

### 3.1. Olfactory Critical Period in Mice 

The olfactory critical period in mice begins right after birth at postnatal day 0 (P0), when neonates experience their chemical environment for the first time, and lasts until P7 [45]. During this period, stimulus-driven mechanisms are in place to drive plasticity in these circuits. Similar to sensory deprivation experiments in the visual system, unilateral naris occlusion and reopening experiments revealed that when the naris was reopened after P8, both pre and postsynaptic markers in the OB were significantly reduced and their expression did not recover to normal levels. Behaviorally, mice with single naris occlusion between P0 and P10 showed reduced responses to odors and poor odor discrimination abilities when compared to mice without occlusion. Also, when the occluded naris was opened before P6, the mice demonstrated similar odor detection and discrimination capabilities as mice without occlusion [45]. It is interesting to note that this critical period coincides with the period during olfactory circuit development when OSN-M/T cell synapses are refined through Sem7A-PlxnC1-dependent postsynaptic mechanisms [44]. In fact, the expression of Sem7A is dependent upon intrinsic OR activation-dependent OSN activity [87], which imparts a unique level of Sem7A expression in individual glomeruli [45]. During the critical period, when mice are exposed to a specific odor (stimulus over-exposure) during P0–P8, there is an increase in the levels of Sem7A expression in a stimulus-dependent manner. Increased Sem7A expression leads to faster dendrite selection and maturation, and consequently an increase in the volume of the odor-specific glomerulus. The receptor for Sem7a, PlxnC1, is specifically expressed in the M/T cell dendrites during the first week of life and gradually diminishes after P8 [44]. This limits the time frame of the critical period, as mice exposed to similar stimulus protocols after P8 do not exhibit such changes in the cognate glomerulus. It was also observed that, in both Sem7A and PlxnC1 KO mice, postsynaptic density formation and dendrite maturation were severely affected, and they both failed to undergo critical period odor exposure-dependent changes as compared to wild-type mice. Further, PlxnC1 KO mice have defective imprinting memory as evidenced by avoidance of social interactions in the adult PlxnC1 knockout mice. All this evidence points towards a central Sem7a-PlxnC1-dependent mechanism that modulates circuits during the critical period in response to the valence of the sensory stimulus. Inhibitory local interneurons also develop during this postnatal period, and it is yet to be seen if the development of these inhibitory neurons plays a similar role in shaping the critical period as the inhibitory neurons during the critical period in the visual circuit [6,7,8,88]. Therefore, once the initial olfactory circuit develops through specific hereditary instructions, environmental factors interact with gene expression mechanisms later during development to modify functional circuits during the critical period. Thus, this odor experience-dependent plasticity matches all four features of CPP in that it occurs during a specific time window when the olfactory network (including local inhibitory circuitry) is still developing and there are genetic programs in place that enable long-lasting changes in synaptic structure, function, and behavior. 

### 3.2. Olfactory Critical Period in Drosophila 

Relative to mammals, much more is known about critical periods in insects. The olfactory critical period in *Drosophila* (Figure 3) begins upon eclosion of the adult fly from its pupal case and lasts for 48 h. In flies, most studies of critical periods rely on prolonged exposure to odors rather than depriving the olfactory organs (the antennae) of sensory input. This is likely because in dipterans, the vast majority of OSNs project bilaterally to the AL, thus making internal comparisons difficult [69,89]. Initial experiments on the olfactory critical period in flies [15] demonstrated activity-dependent morphological changes in the cognate glomeruli as well as behavioral changes observed a week after odor exposure ended. Briefly, when young flies were exposed to high concentrations of benzaldehyde or isoamyl acetate for 4 days between 2–5 days post eclosion, they showed a reduced behavioral response to the exposed odors as compared to other odors. Furthermore, flies chronically exposed to benzaldehyde showed a marked decrease in the volume of specific glomeruli while the volume of the whole AL remained unchanged. Similar experiments with isoamyl acetate showed a marked decrease in the volume of a different glomerulus [15,16,17]. This was marked by a reduction in the synaptic density of the affected glomeruli, which were dependent on cAMP as both mutants for phosphodiesterase and calmodulin-activated adenylyl cyclase failed to undergo these morphological and behavioral changes [15,17], demonstrating a critical role for cAMP signaling. As observed in the mouse OB, the critical period of olfaction in the fly is also marked by synaptogenesis in the AL, with a 38% increase in volume between 1–12 days post eclosion. This increase in volume is due to unique trends of volume increases in individual glomeruli [17]. However, we do not know if mechanisms analogous to Sem7a-PlxnC1 signaling as observed in the mice olfactory circuit (see Section 3.1) are involved in synaptogenesis in the fly. The initial experiments on CPP in the AL of *Drosophila* were performed before the odor tuning of ORs was defined and OR responses were mapped onto distinct glomeruli [59,61,69,81]. Once the molecular map of odor coding was established, the topic of critical periods was reanalyzed independently in several studies. These studies confirmed that the critical period of olfaction lasts up to 48 h post eclosion [21,24,32] and demonstrated that the morphological and behavioral effects are reversible. The glomerulus responsive to geranyl acetate (GA) was an exception to this rule as it was shown that the GA-induced changes in the cognate glomerulus can occur when odor exposure starts 48 h after eclosion [27,32,90]. Therefore, although the structural plasticity in response to GA is a form of experience-dependent plasticity, it does not meet our criteria for CPP. Perhaps this is related to the ethological relevance of GA, which is found to be present in physiologically active concentrations in fruit [91] Therefore, the high level of plasticity is probably needed to find food sources in the adult fly, making it adaptable to its changing chemical environment. Prolonged exposure to fruit odors in *Drosophila* during the critical period led to reduced PN output [92]. Similar experience-dependent plasticity has been observed in adult foraging honeybees in response to floral odorous compounds, where prolonged exposure to such odors led to a decrease in glomerular volume [93]. Similar kinds of ethologically relevant structural plasticity have also been reported in ants [94].

In *Drosophila*, a single odor can activate either one (private odor) or multiple (public) chemosensory receptors and their cognate glomerulus. Furthermore, at higher concentrations, a private odor may activate multiple non-cognate glomeruli. However, both public and private odors have been tested at high concentrations and each of them was shown to have unique effects on their cognate glomeruli and behavioral responses. An exception to this rule is CO_2_, which activates a single glomerulus even at higher concentrations. Exposure to CO_2_ during the critical period causes an increase in the volume of the cognate glomerulus (Figure 3b). In contrast, a public odor, ethyl butyrate, causes an increase in the volume of two of its cognate glomeruli [21,24,26,32] and a decrease in the volume of another cognate glomerulus [30,32]. Behaviorally, reduced responsiveness was observed to these aversive odors, while physiologically, odor-induced activation of different subsets of inhibitory LNs was shown to inhibit PN activity following critical period odor exposure [21,24]. In comparison, similar experiments with attractive odors led to physiologically contradictory observations. While exposure to some attractive odorants reduced OSN activity and increased PN responses [28], exposure to other attractive odorants improved the sensitivity [23] or response rate of their cognate OSNs [22]. Behaviorally, such an increase in OSN activity led to increased attractiveness in the exposed flies. Collectively, these studies reveal a high degree of heterogeneity in the impact of CPP on olfactory network structure.

Glomerulus-specific volume increase can be attributed to the expansion of the pre-existing neuronal arbors and an increase in the number of PN arborizations (Figure 3b). Both mechanisms are involved during the critical period. Expansion of OSN-PN synapses was shown to be regulated by the Notch–Delta signaling pathway [27,28]. Briefly, Notch is expressed by OSNs in an activity-dependent manner, which in turn activates Delta in PNs that synapse with these OSNs and leads to an increase in glomerular volume through non-canonical mechanisms (Figure 3c). However, the extent of the increase in volume is regulated by canonical Notch mechanisms through feedback from Delta on PNs [28] (Figure 3d). The increase in glomerular volume is also driven by the increase in the number of PN arborizations [32]; however, the number of viable PNs or OSNs remains unchanged following critical period odor exposure. Further, cAMP-dependent mechanisms in a subset of LNs are required for the increase in PN arborizations. Specifically, the knockdown of an adenylyl cyclase encoded by *rutabaga* in an LN subset was sufficient to prevent an increase in PN arborizations following critical period odor exposure. In addition, the expression of an inhibitory form of the transcription factor cAMP response element binding protein (CREB) was also able to prevent such plasticity. These results are consistent with previous observations about the importance of cAMP in glomerular volume changes [15,17]. 

However, it is important to note that the LNs in these studies are, as a population, pan-glomerular; thus, questions remain about the exact mechanisms that impart glomerulus-specific plasticity. Hence, how cAMP-dependent transcription in the LNs leads to structural and physiological changes only in the affected glomeruli is unclear. One line of evidence showed that knocking down *Ataxin 2* (Atx2) and *Drosophila* homolog of the Fragile-X mental retardation protein (dFMR1) in the CO_2_ sensing PNs impaired physiological, behavioral, and structural plasticity during the critical period [25,26]. The proposed mechanism for this posits that both dFMR1 and Atx2 are required for miRNA-dependent translational repression during CPP at the local LN-PN synapse [25]. However, the exact mechanisms by which Atx2 might promote glomerular-specific translational repression to give rise to structural, physiological, and behavioral changes remains unknown. One explanation could be that Atx2 and dFMR1 regulate the expression of calcium-calmodulin-dependent protein kinase II (CAMKII) at synapses because the knockdown of Atx2 and dFMR1 upregulates CAMKII expression [26]. It is interesting to note that CAMKII is a known regulator of plasticity in visual critical periods [6,7,8,88]. 

In addition to their role in the PNs, dFmr1 has been shown to be involved in OSN remodeling during the critical period. OSN-specific knockdown of dFmr1 prevents critical period OSN retraction whereas overexpression of dFmr1 in the VM7 glomerulus enhances OSN retraction when exposed to EB during the critical period [30]. However, optogenetic activation of OSN-specific dFmr1 knockdown flies still showed OSN retraction, which implies that EB-specific activation is required for this type of remodeling in the OSNs [30]. 

In addition to the activity-dependent transcriptional and translational control mechanisms in place, we cannot rule out the role of neurotransmitters like GABA and glutamate in shaping critical periods. Previous studies have shown that silencing glutamatergic LNs could prevent OSN remodeling during the critical period in an NMDAR-independent manner as NMDAR mutants do not show any defects in OSN remodeling [30]. However, OSN-specific knockdown of NMDAR1 and GABA-A showed impaired OSN remodeling [31]. In addition, GABA-A and NMDAR receptors were shown to be required in the PNs for both structural and behavioral plasticity. Thus, glomerulus-specificity may arise from the combination of excitatory and inhibitory interactions that occur within a given glomerulus during odor stimulation.

In conclusion, during the critical period of olfaction, chronic odor exposure induces odor-specific structural, physiological, and behavioral plasticity in the fruit fly. These physiological changes are driven by local changes in the activity of the OSNs, LNs, and PNs within the cognate glomerulus. Several transcriptional and translational mechanisms are in place that induce structural plasticity in the glomerulus by changing the volume and number of processes of these neurons. Apart from these, neurotransmitters like GABA and glutamate shape intercellular signaling mechanisms during the critical period. However, much remains to be deciphered regarding the interdependence of these mechanisms to bring about odor-specific changes observed during the critical period. Another avenue not explored yet is if and how neuromodulators like serotonin, dopamine, and oxytocin modulate the olfactory critical period. 

## 4. Conclusions and Future Directions

The olfactory system is unique in that it is composed of organized neuropil structures that encode distinct odor cues in the early processing stages like the OB in mice and AL in the fly. The olfactory circuit provides an ideal model to study how distinct environmental cues represented at the periphery lead to complex behavior and perception through further processing at the higher centers in the brain. Leveraging such a highly defined circuit to uncover mechanisms of critical period plasticity is therefore highly advantageous for several reasons. First, as the discrete neuropil structures that encode specific odors are predefined at the periphery, it is possible to track morphological, physiological, and behavioral changes to each and every odor in the repertoire back to distinct circuits in the brain that are identifiable across individual animals. Second, the critical period of olfaction in rodents overlaps with the postnatal development of the immature olfactory circuit. During the critical period, while the circuit is still developing, environmental factors can modulate the genetic programming of the circuit as described above in the case of dendrite selection by OSN axons in the mouse OB [44,45]. Hence, the olfactory critical period posits a unique opportunity to study how genetic programs and environmental cues interact to shape brain circuits during development. Finally, the olfactory critical period reviewed in mice and flies thus far matches the features of critical periods observed in other sensory systems in being restricted to a specific time window, guided by the onset of sensory input, and exhibiting high levels of structural and functional plasticity that modifies behavior and perception in adults.

Literature on the olfactory critical period in both mice and fruit flies has mainly focused on the mechanisms at play within the primary olfactory processing centers, i.e., the OB and AL, respectively. However, the behavioral changes may not be solely due to the changes in these primary olfactory centers. Higher centers of the brain that drive behavior may modulate these changes. In fact, functional feedback loops are known to exist between the mouse OB and piriform cortex and basal forebrain regions [95]. In *Drosophila*, feedback loops have been described between the MB and the AL [96]. Although the PNs did not show structural plasticity in their axonal projections to the LH and MB following chronic exposure during the critical period [20], we cannot rule out functional plasticity within other LH or MB neurons. Additionally, it is not known if functional LH or MB circuits are required for the expression of CPP in the AL. In adult mice, the M/T cells, are known to be modulated by feedback from the cortex that helps in odor discrimination, identity, and coding in the OB [97,98,99]. However, as with insects, it is unclear whether these circuits are required for the proper expression of CPP. Such circuits could modulate CPP by regulating the activity of early olfactory circuits during development.

Another unexplored avenue in the olfactory critical period is the role of neuromodulators and hormones. Differential expression of serotonin receptor 2C in the kitten’s visual cortex determines the location and type of plasticity that is induced during the critical period [100,101]. Other studies showed that administering selective serotonin reuptake inhibitors in adults increased serotonin presence in the brain, which led to the reopening of critical periods like ocular dominance plasticity in the visual cortex owing to the reduction in inhibition by GABAergic interneurons [102]. Hence, it will be interesting to see in future work whether the opposite effect holds true, i.e., if reducing serotonin activity could reduce GABA function and delay the maturation of inhibitory circuits and thereby extend the critical period. Like neuromodulators, hormones could also play a role in regulating CPP, as they are known to be involved in the development of olfactory circuits. For example, defective social interactions in oxytocin KO mice are rescued when these mice are administered oxytocin as neonates during the olfactory critical period [44]. Initial experiments examining the branching patterns of PNs in the lateral horn of *Drosophila* showed no visible changes following odor exposure during the critical period [19]. However, we cannot rule out the contribution of the LH and MB in modulating the behavioral changes seen following critical period odor exposure, which may arise due to changes in connectivity or biophysical properties of neurons responsive to the chronically exposed olfactory stimuli. One possible mechanism could be that as activity changes in the PNs, there could be structural changes at the synapses in the MB and LH that give rise to the modified feedback from the MB to the AL.

The studies on *Drosophila* olfactory critical period reviewed above all relied on chronic odor exposure at a high concentration to study the changes induced during the critical period. However, such high concentrations could lead to the activation of multiple glomeruli [59,103] and it is unclear then how such glomerular-specific changes were seen during the critical period. In the AL of flies, lower odor concentration recruits lateral excitation to promote sensitivity in PNs via excitatory LNs (eLNs). At higher odorant concentrations, along with OSNs, these eLNs activate inhibitory GABAergic LNs to induce gain control mechanisms and regulate global AL responses [103]. In the absence of ORN-PN excitation, odorants can still invoke excitatory responses through lateral excitation [104]. Indeed, it was seen that chronic odor exposure at naturally occurring or low concentrations during the critical period induced limited changes in the cognate PNs and differentially affected the activity of surrounding PNs via lateral excitation [33]. Similarly, in mice, chronic exposure to food odor during the olfactory critical period led to differential activation of different M/T cells in the OB [105]. These findings underscore the need to explore how such mechanisms modulate the critical period and if higher/order centers in the brain modulate such lateral activation through feedback loops. 

Further questions remain about the exact mechanisms that impart glomerular/specific plasticity. Neurotransmitters like GABA and glutamate shape intercellular signaling mechanisms during the critical period. However, much remains to be deciphered regarding the interdependence of these mechanisms to bring about the odor-specific changes observed during the critical period. Furthermore, enhancing or reducing the GABA function could modulate the duration and closing of critical periods [7].

The maturation of inhibitory circuits in the visual cortex coincides with the closing of the critical period in the visual circuit. Observations in both mouse and fly olfactory networks also show a clear overlap between the time when the development of inhibitory neurons takes place and the critical period onset and closing. In both organisms, future work needs to confirm whether the closing of the critical period coincides with the maturation of these inhibitory neurons in their respective olfactory circuits. Such experiments will provide an opportunity to understand common principles that are at play during critical periods across sensory modalities. 

## Figures and Tables

**Figure 2 biology-13-00094-f002:**
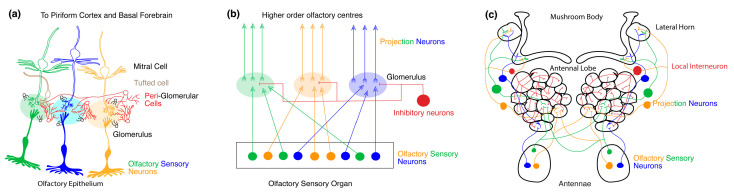
Organization of olfactory processing centers. (**a**) Organization of the primary olfactory system in mice. The olfactory epithelium contains the dendrites and cell bodies of multiple OSNs expressing different odorant receptor proteins (ORs) (depicted by different colors). The axons of all the sensory neurons expressing the same receptor subtype converge to the same glomerulus and synapse with mitral and tufted cells (M/T cells). The glomeruli together constitute the olfactory bulb (OB). Various types of intra and interglomerular inhibitory interneurons also synapse onto each glomerulus. (**b**) Generic organization of the primary olfactory system in mice and fruit flies. Throughout the olfactory organ, dendrites, and cell bodies of OSNs expressing different ORs are distributed, the axons of which synapse onto distinct neuropil structures called glomeruli. Together the glomeruli make up the primary olfactory processing site. Here, projections neurons (PNs) refer to neurons with dendrites within the primary olfactory processing site and axons that project to higher-order olfactory neuropils. Within each glomerulus, inhibitory neurons also synapse with OSNs and PNs. The higher-order olfactory centers receive axonal output from the PNs. (**c**) Organization of the primary olfactory system in Drosophila. In flies, the antennae contain OR-expressing OSNs that project onto distinct glomerular neuropil and synapse with PNs. The glomeruli together constitute the AL in the fruit fly. Within the AL, local inhibitory interneurons also synapse with OSNs and PNs. The PNs project onto higher-order olfactory centers like the mushroom body (MB) and the lateral horn (LH).

**Figure 3 biology-13-00094-f003:**
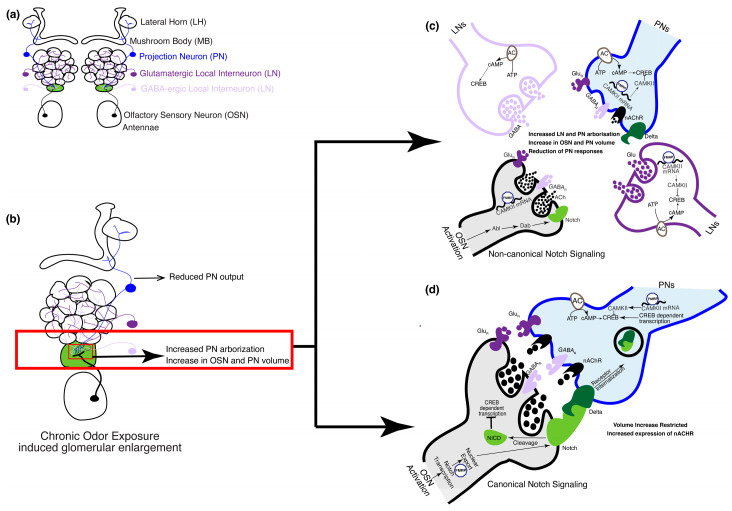
The olfactory circuit underlying critical period plasticity in *Drosophila*. (**a**) The olfactory circuit shows cell types involved during the critical period. In the fruit fly, the olfactory circuit starts at the periphery of the antennae that houses the cell bodies of the OSNs. Discrete AL glomeruli are formed by OSN axons, PN dendrites, and GABAergic and glutamatergic LN processes. (**b**) Upon chronic odor exposure during the critical period, the cognate glomerulus that responds to the odor increases in volume (shown in green). Volume increase of cognate glomeruli is a result of a Notch-dependent increase in the volume of PN and OSN arbors [27,28,90] and an increase in the number of PN processes [32]. (**c**) OSN activation during the critical period following chronic odor exposure activates PNs and LNs. In the OSNs, FMRP upregulates transcription of calcium calmodulin-dependent kinase II (CAMKII) that inhibits the transcription factor cAMP response binding element (CREB). Non-canonical Notch signaling pathways mediate an increase in the volume of the OSN and PN arbors [27,28,90]. Calcium calmodulin-dependent adenylate cyclase mediates CREB-dependent gene transcription in the LNs, which is required for the increase in the number of PN arbors. (**d**) Notch–Delta interaction between OSNs and PNs, respectively, through the canonical Notch signaling pathway, which limits the extent of volume increase of OSN and PN arbors. FMRP aids in the nuclear export of Notch within the OSNs.

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
