# Peer review of "Olfactory Critical Periods: How Odor Exposure Shapes the Developing Brain in Mice and Flies"

_biology, 2024, doi:10.3390/biology13020094_

Round 1

Reviewer 1 Report

Comments and Suggestions for Authors

The authors summarize the literature in mice and fruit flies for the olfactory critical periods during which odor exposure shapes the developing brain, particularly for the olfactory bulb and antennal lobe, respectively. The mechanisms of critical period plasticity mediating odor exposure in the antennal lobe are the changes in the neuronal processes and synapses of OSNs, PNs, and/or LNs. Such changes in the activity of the OSNs, LNs, and/or PNs might be mediated by the responses of the inter- and intra-cellular signals to the environmental factors. The review was well-written and provided novel insight into the examination of neural circuit plasticity across modalities at the specific time window of brain development.

Author Response

We would like to thank the reviewer for their supportive comments about our manuscript. 

Reviewer 2 Report

Comments and Suggestions for Authors

This manuscript reviewed the olfactory critical periods in flies and mice, providing a current understanding of related filed. It presented current understanding in mechanisms underlying olfactory critical periods and put forward remained questions needing future explorations for better understanding of critical periods. This review was organized and written well, clearly focusing on olfactory critical periods, and highlighting its importance in helping understand other types of critical periods. I have no comments and think it can be accepted in present form. 

Author Response

We greatly appreciate and thank the reviewer for their supportive comments about our manuscript. 

Reviewer 3 Report

Comments and Suggestions for Authors

The review by Mallick et al. is devoted to an interesting topic - the critical periods of the olfactory neural networks formation. Structural plasticity, depending on previous experience, is certainly one of the most important issues in neurobiology. The manuscript deserves the attention of readers, but should be revised and improved before publication.

First of all, the references included in the review - of the 113 cited sources, 27 were published before 2000 and another 30 from 2000 to 2010. That is, half of the cited literature is more than 15 years old. The authors need to justify the relevance of the information provided. Reviews of this data probably already exist - what is new about this work?

Other comments and suggestions:

1. Figure 1. The sheme is not clear enough. What is indicated by green and purple? What do ellipses, triangles and rectangles represent?

2. What does the sentense mean: Although the chemosensory receptors in the fruit fly are structurally similar to the mice ORs in that they are seven transmembrane domain proteins, their topology is quite distinct from the rodent GPCRs (lines 137-139). What exactly is the difference?

3. The authors need to describe the mentioned in section 3.2 Sem7A-PlxnC1 signaling pathway in more detail.

In general, the review lacks a section devoted to Mechanisms Regulating Critical Period Plasticity in mice, similar to the section on Drosophila - such a section would have strengthened the manuscript.

There is also a missing Figure illustrating the structure of the olfactory circuit in Mice.

4. Figure 2. Panel 2a lacks the designation AL glomeruli, which is described in the legend. It is not clear what panel b is needed for; all the information given on it is on panel c.

5. What is private and public odor (line 281)? Is this concept applicable to insects  only? Or does a similar classification of odors also true for mammals?

6. It is necessary to organize the abbreviations given in the text. For example, the abbreviation OB is entered on lines 124, 364, 384. Similarly, the abbreviation AL. I also recommend that authors create a list of abbreviations.

7. The reference list is formatted carelessly; it should be formatted in a consistent style.

Author Response

We would like to thank the reviewer for their insightful feedback on our manuscript and greatly appreciate the changes they suggested.  We have addressed the issues as described below in blue text and in the main manuscript where applicable:

The review by Mallick et al. is devoted to an interesting topic - the critical periods of the olfactory neural networks formation. Structural plasticity, depending on previous experience, is certainly one of the most important issues in neurobiology. The manuscript deserves the attention of readers, but should be revised and improved before publication.
First of all, the references included in the review - of the 113 cited sources, 27 were published before 2000 and another 30 from 2000 to 2010. That is, half of the cited literature is more than 15 years old. The authors need to justify the relevance of the information provided. Reviews of this data probably already exist - what is new about this work?

> As described in the final paragraph of our introduction, the purpose of this review is to identify key principles that are common to critical periods of olfaction in two of the commonly studied model systems in vertebrates and invertebrates. Critical period literature in sensory systems have been individually reviewed across various model systems, however we are not aware of reviews that discuss critical periods of the same sensory system across vertebrates and invertebrates. By making comparisons across taxa, we highlight cellular and molecular mechanisms that are fundamental to this important form of plasticity. We cited older literature because critical periods have a rich history in neuroscience and there are important studies that should be highlighted as foundational.

Other comments and suggestions:
1. Figure 1. The scheme is not clear enough. What is indicated by green and purple? What do ellipses, triangles and rectangles represent?

> We have provided greater clarity in the legend for Figure 1 to address this concern. 

2. What does the sentence mean: Although the chemosensory receptors in the fruit fly are structurally similar to the mice ORs in that they are seven transmembrane domain proteins, their topology is quite distinct from the rodent GPCRs (lines 137-139). What exactly is the difference?

> Vertebrate ORs signal using molecular mechanisms that are classically attributed to GPCRs. However, insect ORs are inverted, with their C-terminus facing the extracellular space and each insect OR forms a heterodimer with a broadly expressed co-receptor to create an ion channel. Thus, the molecular mechanisms for transduction are quite different. However, as this is not a critical point to this review, we have removed this sentence to streamline this section. 

3. The authors need to describe the mentioned in section 3.2 Sem7A-PlxnC1 signaling pathway in more detail.

> The Sem7A-PlxnC1 signaling pathway is described in detail in Section 3.1(lines 214 - 233). We refer to this in section 3.2. We have clarified the sentence to indicate this. 

4. In general, the review lacks a section devoted to Mechanisms Regulating Critical Period Plasticity in mice, similar to the section on Drosophila - such a section would have strengthened the manuscript.

> As olfactory critical periods in mice is a nascent field, there is unfortunately very little work that has been done on mechanisms regulating this form of plasticity in the olfactory bulb. What small amount that has been done is described in the section titled “Olfactory Critical Period in Mice” where we describe the studies on Sem-7a/PlxnC1. The work on olfactory critical periods in mice has remained focused upon phenomenological changes rather than the underlying molecular mechanisms. Therefore there has been in totality too little mechanistic literature to merit its own section, so we made a single section discussing both. For this reason, we integrated the mechanistic studies with the phenomenological studies. To keep the manuscript consisted, we decided to merge the mechanistic studies with the phenomenological studies for Drosophila as well in section 3.2.

5. There is also a missing Figure illustrating the structure of the olfactory circuit in Mice.
> Thank you for pointing this out. We have added a new Fig2 to address this. 

6. Figure 2. Panel 2a lacks the designation AL glomeruli, which is described in the legend. It is not clear what panel b is needed for; all the information given on it is on panel c.
> We greatly appreciate this suggestion and have rectified this and removed panel b. 

7. What is private and public odor (line 281)? Is this concept applicable to insects  only? Or does a similar classification of odors also true for mammals? 

> Private odors are those that activate only a single olfactory receptor protein type, whereas public odors activate multiple olfactory receptor protein types. It is difficult to determine if such a concept applies to mammalian olfactory system as the cognate odor ligands remain unknown for so many mammalian olfactory receptor protein types. In other words, there are 100s of mammalian ORs, most of which have not been “de-orphanized” so we cannot make a claim as to whether the concept of public vs. private odors is applicable. We will rephrase the sentence on line 281 to read “In Drosophila, when a single odor  activates only one receptor subtype it is called a private odor, whereas when a single odor that can activate multiple receptor subtypes is  referred to as a public odor.” 

8. It is necessary to organize the abbreviations given in the text. For example, the abbreviation OB is entered on lines 124, 364, 384. Similarly, the abbreviation AL. I also recommend that authors create a list of abbreviations.

> Thank you for this suggestion. We have made the requested change for OB and AL. We have combed through the paper to make sure we use abbreviations for terms used multiple times and have therefore reduced the number of abbreviations we use. Therefore, we feel a table is no longer necessary.        

7. The reference list is formatted carelessly; it should be formatted in a consistent style.

> Thank you for pointing this out. We have edited to maintain a consistent font. The reference list was organized according to the MDPI citation style. This is requested by the journal and outside of our control. 

Round 2

Reviewer 3 Report

Comments and Suggestions for Authors

The authors significantly improved the manuscript and answered all my questions